# Study on the Resistance of a Large Pure Car Truck Carrier with Bulbous Bow and Transom Stern

Xiaoqing Tian [1,2] , Tianwei Xie [2], Zhangming Liu [2], Xianghua Lai [3,*], Huachen Pan [2] , Chizhong Wang [1], Jianxing Leng [1] and M. M. Rahman [2,*]

1   Ocean College, Zhejiang University, Zhoushan 316021, China; tianxiaoqing@hdu.edu.cn (X.T.); cz_wang@zju.edu.cn (C.W.); jxleng@zju.edu.cn (J.L.)
2   School of Mechanical Engineering, Hangzhou Dianzi University, Hangzhou 310018, China; huachen_pan@hdu.edu.cn (H.P.)
3   Yangfan Group Co., Ltd., Zhoushan 316100, China
*   Correspondence: laixianghua@ejianlong.com (X.L.); md.mizan68@gmail.com (M.M.R.)

**Abstract:** The resistance of a large Pure Car Truck Carrier (PCTC) with a bulbous bow and a transom stern is evaluated in the present paper. Several cases at nine different ship speeds in calm water are simulated and results are compared with the experimental measurements. The maximum relative error is 0.93% at a Froude number (*Fr*) of 0.209. The total resistance coefficient of the ship in calm water shows a parabolic trend with increasing Fr, and it reaches a minimum value at *Fr* = 0.1794. Furthermore, the cases of the ship in regular waves with six different wavelengths and three wave heights are simulated. It is observed that the total resistance exhibits a quadratic relationship with the wavelength when the wave height is fixed. The wave-making resistance increases with the increase in wave height at any fixed wavelength, and it reaches a maximum value when the wave-length is 1.2 times the ship length ($L_{pp}$). Additionally, we also investigated the resistance in three different sea states at four different speeds. When the significant wave height of irregular waves is the same as regular waves, the wave-making resistance under irregular waves is much smaller than that of the regular waves. All of these results indicate that the bulbous bow and transom stern can reduce the wave-making and residuary resistances, which can provide a useful reference for the subsequent design and manufacturing of related ships.

**Keywords:** total resistance; CFD; zonal method; wave-making resistances

## 1. Introduction

In recent years, automobile exports from China are on the rise, and consequently, the ship transportation efficiency has received more attention. Ships designed for transporting cars typically have bulbous bows and transom sterns to assist in reducing both the wave-making and residuary resistances.

The artefact of reducing the resistance adhering to the Pure Car Truck Carrier (PCTC) is a hot topic. However, the Energy Efficiency Design Index (EEDI) of the Marine Environment Protection Committee (MEPC) has been paid more attention in terms of green-house gas emissions from ships [1], which raise a higher requirement for the ship design. Ship resistance is one of the most important factors to determine the fuel consumption [2–4]; hence, an accurate evaluation of the resistance is mandatory.

Computational Fluid Dynamics (CFD) has become the most popular method to determine ship resistance with the fast development of computer technology [5]. An accurate ship resistance predicted by CFD is the most important factor in ship design [6–9]. Based on Michell's theory, Peng [10] presented a numerical method for predicting ship resistance, with an error around 10% when compared with experiments. Saha et al. [11] calculated ship resistance with a speed from 8.0 to 10.0 knots using the commercial software SHIPFLOW, and found it was slightly lower than experimental results. Campbell [12] used Star-CCM+

to simulate ship resistance in confined water. Some in-house codes are also used for ship resistance simulations, and the error is around 5% [5,13]. Moreover, artificial intelligence (AI) was also used for ship resistance predictions, such as a deep neural network model developed by Ao et al. [14], a radial basis function neural network modified by Yang et al. [15], and artificial neural networks developed by Cepowski [16]. However, most of the AI methods require a lot of feeding data, which increase the computational cost and are time consuming.

In order to reveal the total resistance and flow field characteristics of a PCTC with a bulbous bow and a transom stern in calm water and waves, the zonal method for potential flow in the SHIPFLOW solver is employed for the simulations. The total resistance is calculated at nine different ship speeds in calm water, and the results are compared with those of the experiments. Furthermore, resistance under regular waves and irregular waves is also evaluated at different wavelengths, wave heights and different sea states.

## 2. Model and Methods

### 2.1. Geometric Model

This ship (see Figure 1) used in the present study is designed for a pure car carrier. The hull has a bulbous bow and a transom stern. For the tank test, a model is made at a scale of 1:27.27. The principal dimensions of the ship are listed in Table 1, while the test conditions are given in Table 2.

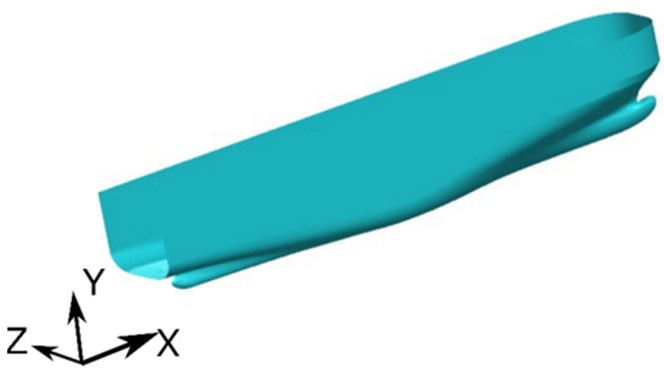

**Figure 1.** The bare hull form of ship.

**Table 1.** Main parameters of full-scale ship and the model.

| Items | Ship | Model |
|---|---|---|
| Length between perpendiculars, $L_{pp}$ (m) | 187.25 | 6.87 |
| Breadth, B (m) | 36.45 | 1.34 |
| Draught, T (m) | 9.5 | 0.35 |
| Wetted surface, S (m$^2$) | 7816 | 10.51 |
| Scale (λ) | 27.27 | 1 |

**Table 2.** Model basin characteristics.

| | |
|---|---|
| Model basin dimensions | 180 m × 10 m × 5 m |
| Density of tank water | 1000 kg/m$^3$ |
| Temp. of the tank water | 14.3 °C |
| Viscosity of tank water at 14.3 °C | $1.16030 \times 10^{-6}$ m$^2$/s |
| Material of model hull | Wood |
| Turbulence stimulation | Carborundum stripes |
| Instrumentation | Electronic dynamometer, load cell, ultrasonic probes |

## 2.2. Ship Resistance

The total resistance $R_{tm}$ of a ship is defined as the force needed to tow the ship at a constant forward speed, and it can be divided into subcomponents in different ways. One way is to divide it into frictional resistance $R_F$ and residuary resistance $R_R$, which includes all components related to the three-dimensional form of the ship and wave-making resistance. It can also be divided into viscous resistance $R_V$ and wave resistance $R_W$ according to physical characteristics. The viscous effect is excluded from the wave resistance, which is therefore considered as an inviscid phenomenon. A boundary layer is created along the entire hull which grows downstream. The thickness of the boundary layer is defined as the distance from the hull surface to the point where the velocity is 99% of the undisturbed flow velocity.

Three types of resistance components are computed in the present project by using the SHIPFLOW [17] solver: wave resistance, frictional resistance and viscous pressure resistance. Wave resistance is obtained from an integration of the potential flow pressure over the body or ship surface. The body surface and free surface are discretized through using the first-order panel and the higher order panel, respectively [18]. The pressure integration is made over the first-order panels, on which the pressure and the normal direction are assumed to be constant for each panel. Integration of the local skin-friction coefficient over the hull surface is carried out to obtain the frictional resistance. The local skin-friction is computed as part of the solution along the stream lines in the boundary layer method and from the wall-shear stress of the wall in the Reynolds Averaged Navier-Stokes (RANS) solution. The potential flow pressure over the stern part of the hull is changed due to viscous effects computed by the RANS solution. The viscous pressure resistance is obtained by integrating the pressure, considering viscous effect. A form factor is computed from the viscous pressure resistance and the frictional resistance where ITTC-1957 [17] line is used as a reference. The frictional resistance coefficient $C_F$ is given by the ITTC-1957 formula:

$$C_F = \frac{0.075}{(logR_e - 2)^2} = \frac{R_F}{1/2SU_m^2} \tag{1}$$

where $Re$ is the Reynolds number and $R_F$ is the frictional resistance, which is the sum of tangential stresses along the wetted surface $S$ area in the direction of the motion, and $U_m$ is the model free-stream velocity. The total resistance coefficient $C_T$ may be defined by Equation (2):

$$C_T = \frac{R_{tm}}{1/2\rho SU_m^2} \tag{2}$$

where $R_{tm}$ is the total resistance.

## 2.3. SHIPFLOW Solver

The SHIPFLOW software uses a Zonal approach to solving the whole flow-field around a ship hull by recurring to the most opportune solver for each single region; the flow-field is subdivided into three zones in order to optimize the numerical computation as shown in Figure 2. In particular, the shape and the potential flow (flow far away from the hull) of free-surface are determined by means of the first-order panel method [19]. The boundary layer on the forward surfaces of the hull is calculated using a boundary layer method, while the flow behind (i.e., in the stern of) the hull is calculated by recurring to a single-phase RANS solver, operating below the free-surface.

It is worth noting that in the potential flow zone, based on the Rankine theory, linear/non-linear free-surface boundary conditions and a higher-order panel method are employed to solve equations, leading to a determination of the wave-making resistance. While the friction resistance is achieved by solving the momentum integral equations in the thin boundary-layer (laminar and turbulent) zone, the viscous pressure resistance is obtained by solving RANS equations with the SST k-ω turbulence model. The computational principle is the XPAN results, which are transferred to the XBOUND and XCHAP.

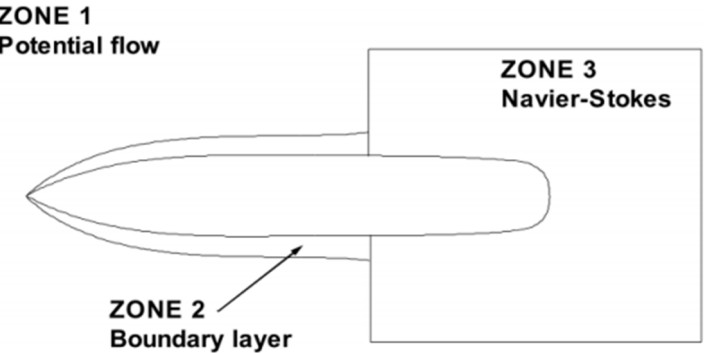

**Figure 2.** Zone partition of the zonal method.

For wave simulations, the nonlinear potential flow solver 'Motions' is used. The Rankine source distribution is used for solving the time-accurate three-dimensional (3D) potential flow in the SHIPFLOW MOTIONS, and the free surface and hull are discretized into quadrilateral panels [17,20].

Since details of numerical methods and formulation of governing equations for incompressible turbulent flows have been extensively documented in the literature, the main features of the potential flow solver used in the SHIPFLOW code are briefly described in the present work. On the assumption of incompressible flow and irrotational motion, the continuity equation becomes:

$$\nabla \cdot U = 0 \tag{3}$$

A velocity potential $\phi$ exists and $U = \nabla\phi = (u, v, w)$, the continuity equation, reduces to the Laplace equation:

$$\nabla^2 \phi = 0 \tag{4}$$

The potential $\phi$ can be decomposed in a double-body potential $\Phi$ and a disturbed potential $\varphi$, representing the effect of free-surface wave:

$$\phi = \Phi + \varphi \tag{5}$$

Boundary condition on the impermeable surface $S$ of the hull is:

$$U \cdot \boldsymbol{n} = \nabla\phi \cdot \boldsymbol{n} = \frac{\partial \phi}{\partial n} = 0 \text{ on S} \tag{6}$$

where $\boldsymbol{n} = n_x \vec{i} + n_y \vec{j} + n_z \vec{k}$ denotes the normal vector to the hull surface in the outward direction. For a generic point P placed at a large distance r from the body, the asymptotic boundary condition is: $\phi(P) = \Phi$ for r $\to \infty$. The kinematic and dynamic boundary conditions are imposed on the free-surface, which is defined as z = $\eta$ (x, y) and can be expressed as:

$$\phi_x \eta_x + \phi_y \eta_y - \phi_z = 0 \text{ on } z = \eta(x, y) \tag{7}$$

Dynamic boundary condition requires that the pressure is constant on the free-surface. Applying the Bernoulli theorem on the undisturbed free-surface far away from the body ($p_\infty$, $U_m$) and at one point on the wavy part of the free-surface ($p$, $U_m$) yields:

$$g\eta + \frac{1}{2}\left(\nabla\phi \cdot \nabla\phi - U_m^2\right) = 0 \text{ on } z = \eta(x, y) \tag{8}$$

where $g$ denotes the acceleration of gravity and is usually taken as a constant. Eliminating $\eta$ from Equations (7) and (8) yields:

$$\frac{1}{2g}\nabla\phi_x(\nabla\phi \cdot \nabla\phi)_x + \frac{1}{2g}\nabla\phi_y(\nabla\phi \cdot \nabla\phi)_y + \phi_z = 0 \text{ on } z = \eta(x, y) \tag{9}$$

The free-surface condition Equation (9) is nonlinear and should be satisfied on the free surface at z = $\eta$ (x, y), which is unknown and can be linearized about the double body solution $\Phi$ by neglecting the non-linear terms of $\varphi$ (i.e., neglecting the velocity derivatives in the z-direction). The linearized free-surface boundary condition may finally be written as:

$$\frac{1}{2g}\left[\begin{array}{c}\phi_x\left(2\Phi_x\phi_x + 2\Phi_y\phi_y - \Phi_x^2 - \Phi_y^2\right)_x \\ \phi_y\left(2\Phi_x\phi_x + 2\Phi_y\phi_y - \Phi_x^2 - \Phi_y^2\right)_y\end{array}\right] + \phi_z = 0 \text{ on } z = 0 \tag{10}$$

using $\varphi = \phi - \Phi$ and $\nabla^2\varphi = 0$. The free-surface boundary condition is applied on the symmetry plane z = 0 for the double-model solution. In addition, a radiation boundary condition has to be imposed to ensure that no waves upstream of the hull will be created, and usually this is enforced numerically.

The solution with the linearized free-surface boundary conditions is obtained through an iterative scheme based on the solution at the previous time step, and it may start either from the undisturbed flow or from a flow assuming the free-surface to be flat. In the first iteration the conditions are applied on the undisturbed surface, while in the later ones they are applied to the wavy surface from the previous iteration. If the process converges, the difference between two subsequent solutions for $\phi$ and $\eta$ tend to be zero. The problem is solved by discretizing the hull surface and the free-surface by quadrilateral panels. On each panel, sources are continuously distributed. More details can be found in [18].

The pressure on the hull surface can be evaluated from the perturbation potential by using a linearized version of the Bernoulli Equation [20]:

$$p + \rho g\eta + \frac{1}{2}\rho \nabla\varphi \cdot \nabla\varphi = p_\infty + \frac{1}{2}\rho U_m^2 \tag{11}$$

and the pressure coefficient can be obtained by Equation (12):

$$\begin{aligned} C_P = \frac{p - p_\infty}{1/2\rho U_m^2} &= 1 - \frac{1}{U_m^2}[2g\eta + \nabla(\phi - \Phi) \cdot \nabla(\phi - \Phi)] \\ &= 1 - \frac{1}{U_m^2}\left[\begin{array}{c}2g\eta + \Phi_x^2 + \Phi_y^2 + \Phi_z^2 - \\ 2\left(\Phi_x\phi_x + \Phi_y\phi_y + \Phi_z\phi_z\right)\end{array}\right] \end{aligned} \tag{12}$$

The wave-making resistance coefficient can be determined by:

$$C_w = -\frac{R_W}{1/2SU_m^2}\iint C_p n_x dS \tag{13}$$

where $R_W$ is the wave-making resistance, $dS$ is the area of the hull surface panel, and $n_x$ is the x-component of unit normal on a surface panel. The wave profile can be obtained from the linearized dynamic free-surface boundary condition:

$$\eta(x,\ y) = \frac{1}{2g}\left[\begin{array}{c}U_m^2 + \Phi_x^2 + \Phi_y^2 - \\ 2\left(\Phi_x\phi_x + \Phi_y\phi_y\right)\end{array}\right] \tag{14}$$

It should be mentioned that the constant panel (first panel) is used to evaluate potential flow, and the velocity and pressure derivatives are computed by an analytical method [18] in the SHIPFLOW solver.

### 2.4. Grid Generation

Grid generation is essential for the numerical simulation. We now load the 3D geometric model above to the SHIPFLOW solver. According to the hull structure and flow characteristics, the XGRID is set as 'fine' and the 'ytarget' is 2.5. The XMESH is divided into the following four parts: bow, hull, upper part of the stern and lower part of the stern. The point numbers for each part in the Z-axis are set as 100, 134, 82 and 40, respectively. They are set as 34 and 16, respectively, in the *X*-axis for the bow and lower part of the stern.

The meshes are automatically generated in the *X*-axis for the hull and upper part of the stern. The maximum flow iterations are set as 20 in the XPAN. The grids are then generated in the SHIPFLOW solver automatically [17] (see Figure 3). The total number of grid points is $1.33 \times 10^6$.

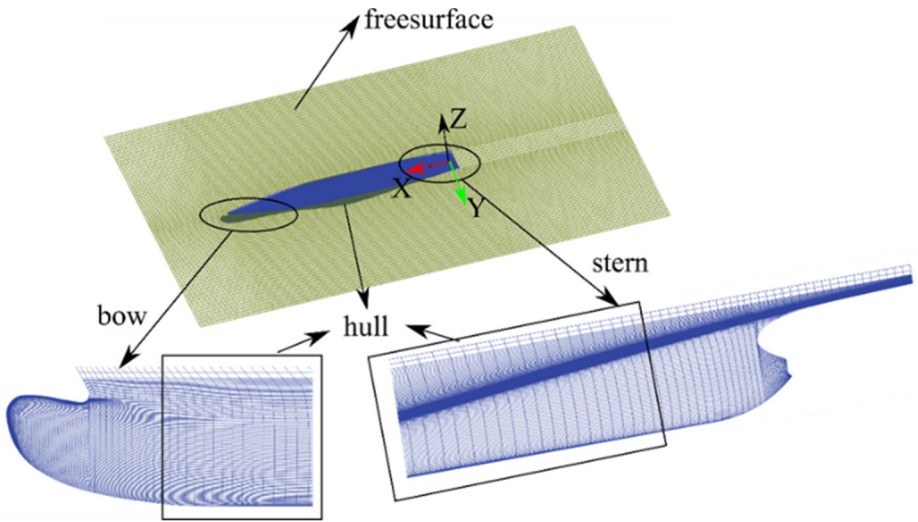

**Figure 3.** Hull and free-surface grid.

### 2.5. Boundary Conditions

For the situation in calm water, '$v_S$' is the ship speed which accounts for the Froude number *Fr*, and the Reynolds number *Re* is also set for the 'offset' geography of the hull. The length between perpendiculars $L_{pp}$ and the draught are set to 187.25 and 9.5, respectively. Iteration counts are set to 20. The sea states from 3 to 5 are set in the 'MOTIONS'.

### 2.6. Experimental Measurements

The model experimental measurements are conducted in a Vienna Model Basin towing tank with the Force Technology. Dimensions of 240 m × 12 m × 5.5 m, ascribed to the towing tank model with an attachment to the Planar Motion Mechanism are used. The depth-to-draught ratio is 17.96, and the pipe water temperature is 14.3 °C. During the tests, the model is freed to have heave and pitch motions in the vertical plane, but otherwise remains constrained. The roughness of the model ship is ignored. The resistance for each point is measured more than 3 times. The detailed information can be found in Ref. [21].

### 3. Results and Discussion

### 3.1. Total Resistance in Calm Water

According to Equation (2), the total resistance $R_{tm}$ is calculated by the following formula:

$$R_{tm} = \frac{1}{2}\rho S U_m^2 C_T \tag{15}$$

The results of total resistance versus the dimensionless ship speed or Froude number *Fr* by the SHIPFLOW solver at different speeds are provided in Figure 4. *Fr* is defined as:

$$F_r = v_S / \sqrt{gL_{pp}} \tag{16}$$

where $v_S$ is the ship velocity.

The values of total resistance are nondimensionalized with the total resistance coefficient formula. The simulation is made with a range of Froude numbers from 0.174 to 0.232, which correspond to speeds $U_m$ = 15 knots to 20 knots, respectively. The experimental results are also provided for comparison in the figure. It is shown that the numerical results by the SHIPFLOW solver are generally in good agreement with the experiments. The

minimum and maximum relative errors are 0.22% and 0.93%, respectively, and the average relative error is 0.62%. Correspondingly, the calculated total resistance by the SHIPFLOW solver is very close to the measured data.

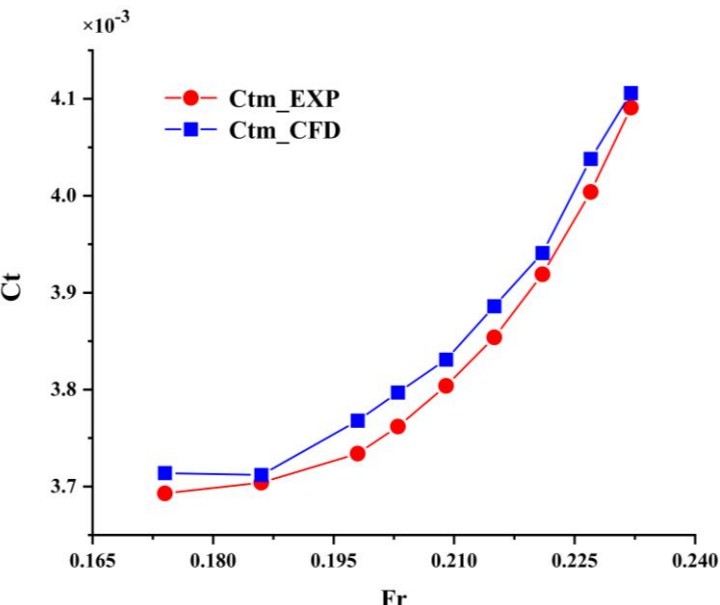

**Figure 4.** Comparison of total resistance coefficients between the simulation and the experimental results.

It should be mentioned that the grids can be generated automatically by the SHIPFLOW solver but still need to be modified by the XMESH and XGRID. The connection between the sub-grids in different domains should be handled carefully. Otherwise, the results will be divergent. The simulation is performed in the processor of Core-i7-6500 and iterations take 1.2 h.

In the simulation, a four-point upwind operator is employed to calculate the velocity derivatives on the free-surface, and an analytical expression is also used for the derivatives at the same time. A small upstream shift of the free-surface collocation points is used to further enforce the radiation condition. The lift force is introduced as a dipole distribution on the lifting surface, and the trailing wake goes together with a flow tangency condition at the trailing edge of the lifting surfaces.

It can be seen that the total resistance coefficient reaches a minimum at $Fr = 0.1794$ and afterward, increases with increasing $Fr$, which grows in a quadratic functional form of $C_t = 141.81 \times Fr^2 - 50.891 \times Fr + 8.2765$, $R^2 = 0.9971$, where $R^2$ is the coefficient of regression, representing the goodness of fit of a model.

In order to understand the variation in wave, three Froude numbers ($Fr = 0.174$, 0.209 and 0.232) are chosen; the results are plotted in Figure 5 for analyzing the overall performance in predicting the wave shapes. From Figure 5a–c, free surface waves and their differences can be clearly seen. Furthermore, it can also be observed that the waves at the bow and stern become more and more pronounced with the increase in $Fr$.

The highest wave peak appears at the bow position, and the wave height shows a strong linear increase with increasing speed, that is, wave-height/$L_{pp} = 33.336 \times Fr - 5.0374$ ($R^2 = 0.9785$). The location of the largest wave is closer to the stern and moves linearly away from the stern as the speed increases; the locations can be expressed by x/$L_{pp} = 2.7967 \times Fr + 0.7895$ ($R^2 = 0.9974$), where the wave height linearly increases as the speed increases, and the formula is: wave-height/$L_{pp} = 5.2767 \times Fr + 0.0371$ ($R^2 = 0.8872$). It can be seen from Figure 5 that the maximum wave height at the stern has a significant downward trend when compared with the wave height near the bow. The drop in the wave height and the change in dimensionless ship speeds form a quadratic curve with a decreasing

ratio = $-60.545 \times Fr^2 + 34.864 \times Fr - 4.2677$ ($R^2 = 0.9878$), indicating that the design of the square stern can significantly reduce the intensity of waves formed by the stern, when *Fr* changes from 0.186 to 0.3898, and it works the best at *Fr* = 0.2879 (with a ship speed of 23 knots), after which the attenuation of the stern wave decreases. However, the wave on the stern of the ship has an enhanced effect when *Fr* is lower than 0.176.

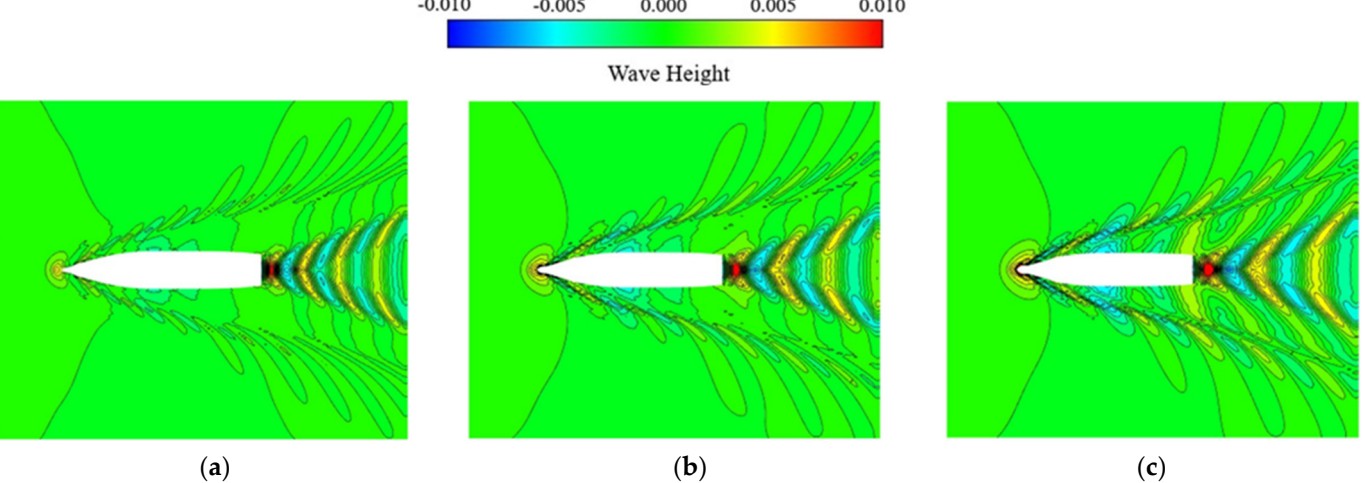

**Figure 5.** Wave distributions around the hull-in calm water: (**a**) *Fr* = 0.174; (**b**) *Fr* = 0.209; (**c**) *Fr* = 0.232.

### 3.2. Ship Resistance under Regular Waves

The speed of the target ship is selected as 17 knots (*Fr* = 0.198), the respective wave heights are chosen to be 1.5 m, 2.5 m and 4.0 m (0.0067, 0.0134 and 0.214 times $L_{pp}$, respectively). The wavelengths (λ) are 0.2, 0.6, 1, 1.2, 1.5 and 1.8 times $L_{pp}$, respectively. The calculated total resistance is shown in Figure 6.

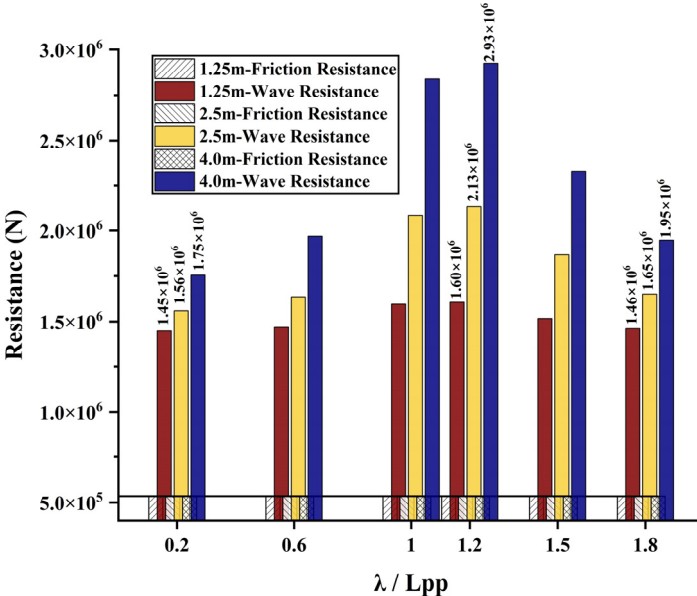

**Figure 6.** The resistance distributions under different conditions.

Figure 6 shows that the frictional resistance of the ship under various working conditions remains basically unchanged, i.e., the frictional resistance has no direct relationship with the wave height and wavelength. Under the condition with the same wavelength to each other, the wave-making resistance greatly increases with the increase in wave

height. When the wave height increases by keeping the wavelength fixed, the increased amplitudes of the wave-making resistance between the adjacent wavelengths become increasingly larger.

Under the three wave heights, the distribution of the total resistance is shown in Figure 7, which is basically a normal distribution. The resistance peaks under the three wave heights are generally at the wavelength of 1.2 times $L_{pp}$.

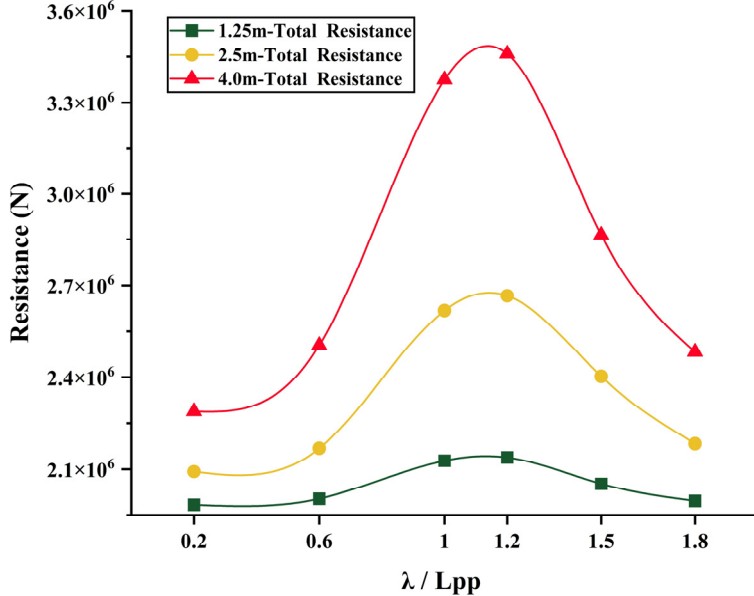

**Figure 7.** Total resistance of the ship under different wave heights and wavelengths.

Figure 8 gives the wave profiles at a wave height of 0.0134 times $L_{pp}$ (2.5 m) with different wavelengths. As the wavelength increases, the maximum wave height will gradually increase. When the wavelength reaches $1.2 \times L_{pp}$, the wave height reaches a maximum value of 3.9 m. After that, as the wavelength increases, the maximum wave height remains unchanged. Furthermore, the draft of the ship gradually increases as the wavelength increases, and it reaches its maximum value when the wavelength is equal to $L_{pp}$, and then the draft of the ship gradually decreases. The maximum wave height is mainly distributed near the bow of the ship, and its largest value is at the bow when the wavelength is $1.2 \times L_{pp}$; the wave-making resistance is the largest at this time, which will probably stay in a resonant situation.

### 3.3. Ship Resistance in Irregular Waves under Three Different Sea States

The sea state of levels 0–2 belongs to the smaller amplitude wave condition, which has almost no effect on the navigation of transport ships; when the sea state level is above level 6, it is not recommended for transport ships to carry out operations at this time. It has been proposed to select 4 typical speeds ($Fr = 0.174$, 0.198, 0.215 and 0.232) of the target ship, and to numerically calculate the resistance suffered by the actual scale state when sailing undersea conditions of 3–5 (ITTC Irregular waves). The calculated total resistance and the wave-making resistance are shown in Figure 9. It can be seen from the figure that the wave-making resistance of the ship also increases linearly with the increase in speed, following the formula $R_w = 5 \times 10^6 \times Fr - 502{,}813$ ($R^2 = 0.9969$). The wave-making resistance basically changes in a parabolic form with the increase in sea state level, and the minimum value of the wave-making resistance occurs when the sea state is between levels 1 and 2.

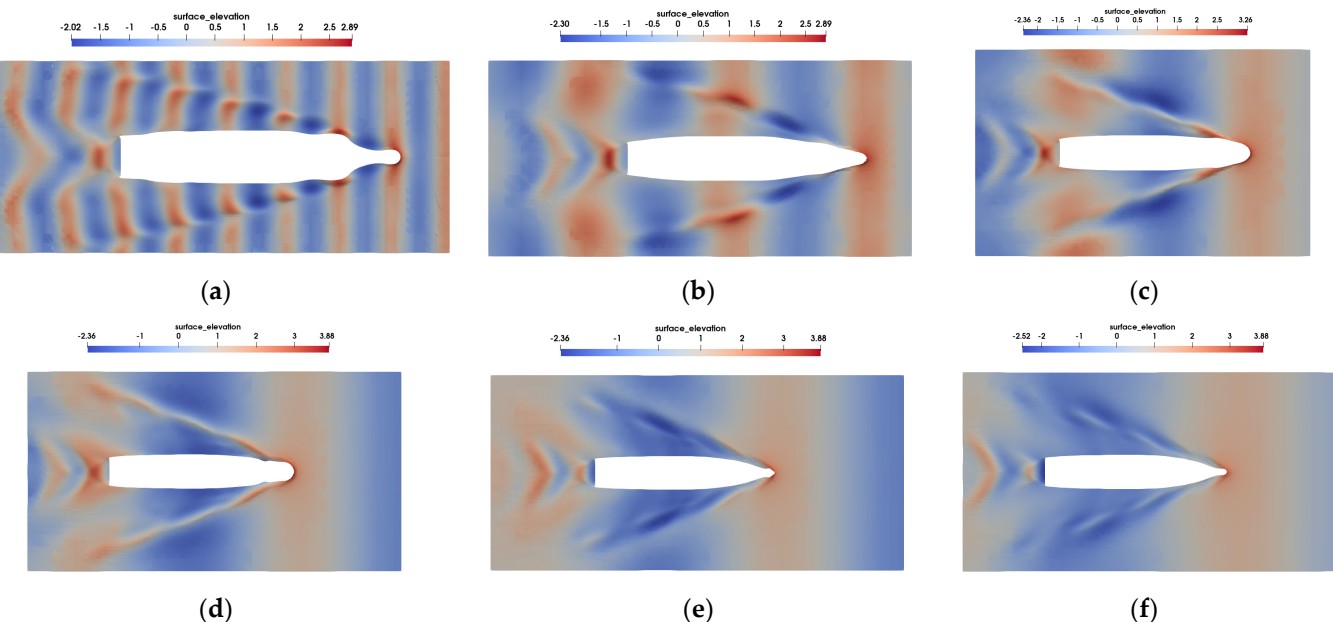

**Figure 8.** At a wave-height of 2.5 m, the wave distributions around the ship are: (**a**) wave-length $0.2 \times L_{pp}$; (**b**) wave-length $0.6 \times L_{pp}$; (**c**) wave-length $1.0 \times L_{pp}$; (**d**) wave-length $1.2 \times L_{pp}$; (**e**) wave-length $1.5 \times L_{pp}$; (**f**) wave-length $1.8 \times L_{pp}$.

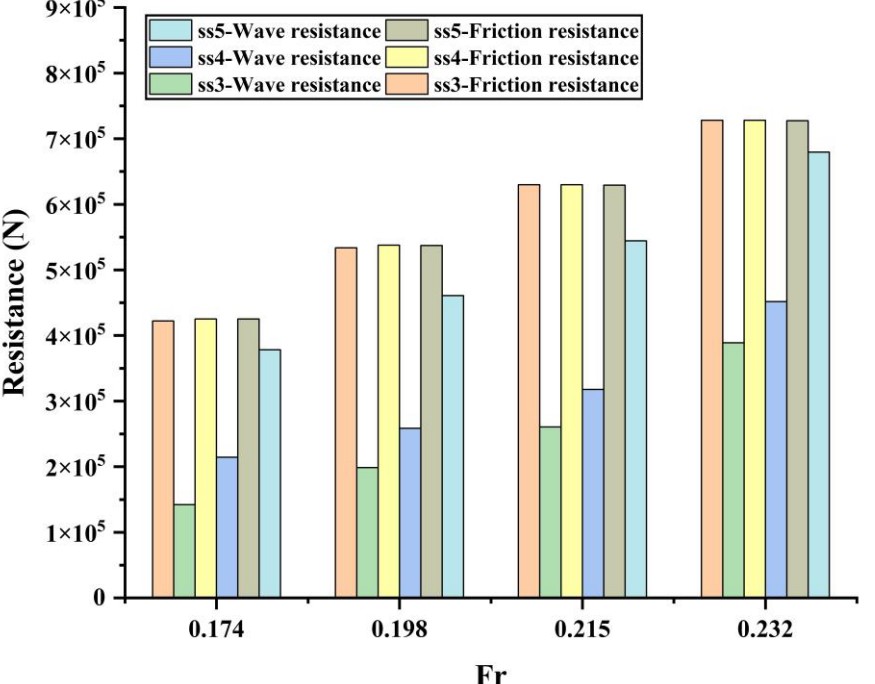

**Figure 9.** The resistance of the ship under different sea states (resistance includes total resistance, friction resistances and wave-making resistance; the nondimensional sailing speed of the ship [*Fr*] is 0.174, 0.198, 0.215 and 0.232; sea states are 3, 4 and 5).

We now make a comparison between Figures 6 and 9. When the significant wave height of an irregular wave is the same as the regular wave height, the wave-making resistances under irregular waves are much smaller. Comparisons of the wave-making resistances are also made between irregular waves and calm water. It is shown that the wave-making resistances under irregular waves are much larger than those of calm water under the same dimensionless sailing speed. This is due to the fact that the irregular wave

is composed of 40 regular waves with different wavelengths and heights. The wavelength of these regular waves falls in a quadratic functional form of the regular wave's list. The longest wavelength increases with the increase in the sea state level, and the shortest wavelength is zero. The significant wave height of the irregular wave is the average of the highest and the lowest wave height.

The wave distributions on the free surface are shown in Figure 10 when *Fr* = 0.198 (sailing speed is 17 knots) and the sea state is 4. The highest wave height in the free surface is 2.4 m. The highest wave is located near the stern. The draught of the bow is slightly shallower than that of the designed situation. Therefore, the resistance of the ship decreases, indicating that the bow can reduce the wave-making resistance when the ship is under the wave scenario.

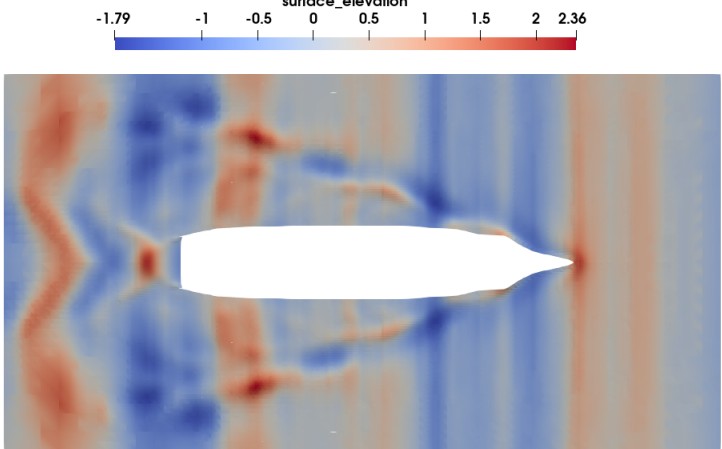

**Figure 10.** The wave distributions on the free surfaces (*Fr* = 0.198, sea state 4).

## 4. Conclusions

A numerical model based on the SHIPFLOW solver is used reliably to simulate the resistance of a PCTC. For the present ship model in calm water, the average relative error for the SHIPFLOW solver is 0.62% when compared with measured data. The computational speed of the SHIPFLOW solver is considerably faster and does not need much computing resources.

Under calm water, the total resistance coefficient of the ship grows in a quadratic functional form ($C_t = 141.81 \times Fr^2 - 50.891 \times Fr + 8.2765$, $R^2 = 0.9971$); the optimum *Fr* is 0.1794. The highest wave peak is around the bow, and the wave height increases linearly with the increase in sailing speed.

The friction resistance has no relationship with the waves. For regular waves, the wave-making resistance increases as the wavelength increases, and the wavelength of $1.2 \times L_{pp}$ is the turning point; after that, the resistance will decrease. For a normal sea state level, basically, the wave-making resistance increases linearly as the level of the sea state increases. The wave-making resistance of irregular waves is much smaller than that of regular waves.

The wave-making resistance can be reduced by a bulbous bow and transom stern in some situations. The bow is sensitive to waves, and it can help the ship to maintain optimum sailing situations. When *Fr* is greater than 0.186 and less than 0.3898, the stern can reduce the trailing waves.

It should be mentioned that the ship model has no accelerated motion in the present simulation of resistance. In an actual navigating environment, the ship motion is usually in six degrees of freedom with acceleration. In such cases, the resistance and motions should be obtained through solving the Kelvin–Kirchhoff equation [22,23] without or with vortex motions due to the viscous effect, which will be considered in future works.

**Author Contributions:** X.T.: Conceptualization, Methodology, Validation, Formal Analysis, Investigation, Writing—Original Draft, Writing—Review & Editing, Project Administration, Funding Acquisition. T.X.: Conceptualization, Methodology, Formal Analysis, Investigation, Writing—Original Draft, Writing—Review & Editing. Z.L.: Conceptualization, Methodology, Formal Analysis, Writing—Original Draft, Writing review & editing. X.L.: Conceptualization, Methodology, Formal Analysis, Investigation, Writing—Original Draft, Writing—Review & Editing. H.P.: Conceptualization, Methodology, Formal Analysis, Investigation, Writing—Review & Editing, Resources, Visualization, Project Administration, Funding Acquisition. C.W.: Conceptualization, Methodology, Writing—Review & Editing, Project Administration, Funding Acquisition. J.L.: Conceptualization, Methodology, Validation, Investigation, Writing—Review & Editing. M.M.R.: Conceptualization, Methodology, Investigation, Writing—Review & Editing. All authors have read and agreed to the published version of the manuscript.

**Funding:** The research was supported by the Key R&D Program of Zhejiang Province (2018C04002; 2021C03013), the Fundamental Research Funds for the Provincial Universities of Zhejiang (No. GK229909299001-007) and National Natural Science Foundation of China (No. 52171325), to which the authors are most grateful.

**Institutional Review Board Statement:** Not applicable.

**Informed Consent Statement:** Not applicable.

**Data Availability Statement:** Not applicable.

**Conflicts of Interest:** The authors declare no conflict of interest.

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
