# Peer review of "Study on the Resistance of a Large Pure Car Truck Carrier with Bulbous Bow and Transom Stern"

_jmse, doi:10.3390/jmse11101932_

Round 1

Reviewer 1 Report

This paper has a good work but also many things to improve. However, I think that the most important issue is the lack of scientific soundness. I mean, the study developed here is just CFD simulations of a PCTC. In other words, if I start to simulate different ship lines in calm water and waves, can this be considered research? Also, the conclusions are in some ways obvious and can not be focused on a code validation.

I think the authors need to think and work in a deeper research area before considering this paper for publication.

Author Response

  • Thanks for the review. We focus on the hydrodynamic characteristic of a PCTC. It is not a fundamental research. However, it may have important application value in ship engineering.
  • Although the simulation is for a PCTC, the method used in the simulation can be utilized for other ship with different ship lines in calm water and waves through setting ‘Xgrids’ and ‘Xmesh’ again in SHIPFLOW (see Section ‘2.4 Grid Generation’).
  • We made revision on the conclusion such as deleting “SHIPFLOW can be used faithfully to simulate the resistance of a ship” and focus is on what we found in the simulation.

Reviewer 2 Report

I think that the presented work should be an interesting development in the field of hydrodynamical engineering under investigation with application to the problem of stability of the big ship’s motion in the open aquatory or sea. So, this scientific work is original and results, presented herein, are novel; besides, this work corresponds enough to the scope of the journal and to the broad readership of the journal.
Thus, this Reviewer will approve the presented research after minor corrections (mainly, in the list of literature).

Meanwhile, I recommend for authors to mention additionally the remark (in Conclusion section) regarding taking into account the hydrodynamical torques arising on the surface of "fluid-body interactions" when it moves through the surrounding hydrodynamical medium, including those stemming from both laminar and turbulent regimes in a sideway direction, for futher investigating stability in a future works:

1) Miloh T., Landweber L. (1981). Generalization of the Kelvin-Kirchhoff equations for the motion of a body through a fluid, Physics of Fluids, Vol. 24, Issue 1, Pages 6-9. 

2) Ershkov S.V., Christianto V., Shamin R.V., Giniyatullin A.R. (2020). About analytical ansatz to the solving procedure for Kelvin–Kirchhoff equations. European Journal of Mechanics, B/Fluids, vol. 79C, January–February 2020, pp. 87–91.

Also I recommend to mention there regarding the fact that the magnitude of acceleration due to gravity is known to be varying on the surface of the Earth depending on the location (not less than ~ 0.5%, but taking into account also the tidal effects along with the additional spherical harmonics from gravitational interaction "Sun-Earth-Moon", not less than ~ 3%). Please, comment this possible variability of the acceleration due to gravity g participating in Eqns. (8)-(9), (11)-(12), (14), etc. (this may slightly influence on the hydrodynamical stability in result).

 List of minor remarks:

1. In all equations, we can see absence of the sign of dividing numerator on denominator (please, fix this technical issue in formulae);

2. Please, clarify terminology "clam water" for the common readers of the journal (what does it mean clearly). May be, you meant "calm water"? In this case, recheck all the text (beginning from Abstract); I meet right denotation "in calm water" only when reading Conclusion (line 304), in all other parts of text there is denotation "clam water" (including in Conclusion further, line 307);

3. In 1st sentence in Introduction, please change "trans-ported" to "transported";

4. Line 106 (page 3): please, change "It is worth nothing" to "It is worth noting";

5. Define what does it mean "g" in Eqn. (8), there is no denotation for "g";

6. Line 146: please, change in "The solution is prosecuted" the word "prosecuted" to another, it sounds not good in English;

7. Line 169: The dot is required before "They" in the middle of line 169;

8. Denotations are unclear: in line 177 do you mean "v_s" (where "s" is low index) instead of "vship" (?); also, denotation for the Froude number (in lines 178, 186) is Fr, while in formula (15) its denotation is F_r (where "r" is low index);

9. Please, change denotation "kn" in line 190 to "knots" for the common readers of the journal (or clarify this abbreviation); also, in lines 231, 294;

10. Line 209: Please, delete "he" from "the he total resistance" or clarify what does it mean;

11. Please, correct the year from 2023 to 2021 in your ref. 14;

12. Please, update the list of references with respect to recent works published in the field under investigation in 2022-2023 years (I saw only 1 work of 2022 year).

Simple logic, algebraic manipulations, numerical estimations and conclusions are under responsibility of the authors. My recommendation: minor revision (mainly, in the list of literature). I wish to review this article after revision again (the list of minor questions or notes for corrections could be suggested for authors additionally if manuscript will pass 1-st round of review successfully).

Author Response

Thanks for the review. In the field of ship hydrodynamics based on both potential and viscous flow theories, All control equations and boundary conditions are derived based on the assumption of a constant acceleration of gravity g=9.81m/s2 and hence it is difficult for us to evaluate the flow field characteristic and magnitude of resistance when considering a variational gravity and tidal effects. We add a sentence “where g denotes the acceleration of gravity and is usually taken as a constant.” to the paper (see Line 144 in revised version).

The other questions in the attachments.

Reviewer 3 Report

General

The authors use the software SHIPFLOW to determine the resistance characteristics of a large Pure Car Truck Carrier (PCTC) in calm water and under regular and irregular waves. Content of the manuscript is good; however, major language and presentation revisions must be done prior to publication. Several examples (but not all) concerning the language and presentation problems are given below to guide the authors.

Specific Points

1.     Title should better be “Study on the resistance of a large Pure Car Truck Carrier with Bulbous Bow and Transom Stern”. Use of acronym in the title should be avoided as it hinders understanding the subject of the paper.

2.     Abstract Lines 11-12: “Several … compared with the experimental.” must be “Several … compared with experimental measurements.”

3.     Abstract Lines 19-20: “When the significant wave height of irregular wave (ITTC) is the same as regular wave, the wave-making resistance …” What is the reason for writing (ITTC) here? ITTC Formula is for frictional resistance and it has nothing to do with waves and wave-making resistance.

4.     Introduction Lines 26-28: “In recent years, automobile export of China is clearly increasing and large numbers of automobiles are trans-ported to other countries. The transportation efficiency has received more and more attentions. Usually, the ship is designed for … ”. This sentence would be much better when rephrased like “In recent years automobile exports of China are on the rise and consequently ship transportation efficiency receives more attention. Ships designed for transporting cars typically have bulbous bows and transom sterns to help to reduce both the wave-making and residuary resistance.”

5.     Table 1: Title should be changes as “Main parameters for full-scale ship and her model.” In the table “Items” should be “Main parameters”. Table 2: Change the title to “Model basin characteristics”

6.     All the manuscript must be completely checked for its language and presentation as exemplified above. For instance, Subsection 2.2 does not give good enough information concerning the resistance. A very obvious missing part is about the experimental measurements. There is no subsection to cover the experiments, who conducted them (were they a part of this work or taken from another group?), etc. According to the extent of work done by the authors a section or subsection must be designated to experiments.

7.     Figure 6: Legend for horizontal axis (wavelength/λ) is clearly wrong. λ is wavelength. (Incidentally, wavelength is written as a single word not as wave length.) The correct legend is probably λ/Lpp. Check all the figures for possible similar errors.

8.     Conclusions: It is better if given without numbering. The first item begins with the sentence “According to the requirements of MEPC, the resistance needs to be optimized.” This is not a “conclusion”, it is a statement of a requirement; therefore, it must be removed. In the next sentence “faithfully” is a wrong word, “reliably” or something similar should be used.

9.     Check the entire manuscript thoroughly for language and presentation problems.

Conclusion

According to this reviewer, after doing extensive revisions to the entire manuscript, it should be reconsidered for publication.

Detailed comments concerning language are given under the overall comments.

Round 2

Reviewer 1 Report

I still think the same. I have read the paper several times and although I find some improvements I cannot see the scientific soundness. Indeed the authors confirm this in their reply "It is not a fundamental research"

They say that this procedure can be applied to other cases and indeed it is good. However, that is the first step of a new research the validation and verification (according with ITTC).

If you read many research, in naval hydrodynamic and even using this code, they start proving what it is presented here but the add some novelty or anything new in ship resistance, waves ... This is not the case. Only a validation is done.

That is the reason why I do not think that this paper should be accepted.

Author Response

Thanks for the review. Please open the attachment.

Reviewer 3 Report

The authors have complied with most of the revision requirements of this reviewer. After going over for a final check on the language for minor problems (such as Page 6 Line 194: "... measurements are done in ..." should be "... measurements were done in ...") the manuscript should be acceptable.

The required revisions of this reviewer were mostly done; therefore, the manuscript may be published.

Author Response

Done
